# Enhancing Water Infiltration through Heavy Soils with Sand-Ditch Technique

**Majed Abu-Zreig** [1,2,*], **Haruyuki Fujimaki** [2] **and Mohamed Ahmed Abd Elbasit** [3,4]

1    Civil Engineering Department, Jordan University of Science and Technology, Irbid 22110, Jordan
2    International Platform for Dryland Research and Education, Arid Land Research Center, Tottori University, Tottori 680-0001, Japan; fujimaki@tottori-u.ac.jp
3    Agricultural Research Council- Soil, Climate and Water, Pretoria 0001, South Africa; MohamedAhmedM@arc.agric.za
4    School of Geography, Archaeology, and Environmental Studies, University of the Witwatersrand, Johannesburg 2000, South Africa
*    Correspondence: majed@just.edu.jo; Tel.: +962-795710134

**Abstract:** Enhancing rainwater infiltration into heavy soils is an important strategy in arid regions to increase soil water storage and meet crop water demand. In such soils, water infiltration and deep percolation can be enhanced by constructing deep ditches filled with permeable materials, such as sand. Laboratory experiments were conducted to examine the effect of sand ditch installed across the slope of a soil box, $50 \times 20 \times 20$ cm$^3$, on runoff interception and water infiltration of clay soil packed at two bulk densities, 1240 and 1510 kg/m$^3$. The experiments were carried out under laboratory conditions using simulated steady flow of about 20 cm/h for a duration of 60 min. Results showed that sand ditches highly reduced runoff and largely enhanced water infiltration into soils. In low-density soil, the average runoff was 15% of inflow volume but reduced to zero in the presence of sand ditches thus increasing soil water storage by 15%. In high-density soil, the presence of sand ditches was more effective; infiltration volume increased by 156% compared to control. The WASH_2D model was used to simulate water flow in the presence of sand ditches; it showed to increase water infiltration and soil-moisture storage thus improving crop production in drylands.

**Keywords:** deep percolation; drought; drylands; rainfed agriculture; runoff harvesting; soil moisture storage

---

## 1. Introduction

Jordan is a country with limited water resources and chronic water shortages. The Kingdom relies mainly on rainwater, but only 1.1% of its total area receives an average of 400–600 mm/year, and 93% is situated in arid areas with an annual rainfall average of 50–200 mm [1]. In its effort to reduce deficits between supply and demand in the agricultural sector, the government of Jordan has been promoting the use of efficient irrigation systems in the Jordan valley and adopting several soil conservations measures in the rainfed-high lands [1]. Floodwater harvesting is also being encouraged in the eastern Badia region. Nevertheless, government must continue to promote and apply best water management options to mitigate the significant imbalance between limited supplies and growing demands.

Rainfed agriculture is a production system where crops are entirely dependent on the amount of rainwater stored in the soil profile [2]. Rainfed production utilizes the thick alluvial soils, which allow high retention storage during the wet season and vegetation subsequently extracts soil water in the next dry season. The system is thus highly vulnerable to rainfall fluctuations, weather changes and soil surface conditions that affect soil moisture storage. Currently, the rainfed agricultural area in Jordan is

estimated at 170,000 ha out of 400,000 ha, but 50 years ago, agricultural production was completely rainfed. Therefore, methods to improve rainfed production and reduce the associated risk must be adopted, such as rainwater harvesting.

The basic principle of water harvesting is to capture precipitation falling in one area and to transfer runoff to a planting area thereby increasing the amount of water available to plants [3]. On-farm microcatchment water harvesting technique in one of the most popular methods used to collect runoff water from runoff area to planting area for crop use [4–7] and was used by Nabateans almost 2000 years ago [8]. They used very sophisticated water harvesting to farm more than 700,000 acres in areas that only get 3–4 in/year of rain. A microcatchment is a specially contoured area with slopes and berms designed to increase runoff from rain and concentrate it in a planting basin where it infiltrates and is effectively "stored" in the soil profile. The collected rainwater is stored in the soil profile to meet the crop water demand during the summer months. The Ministry of Agriculture launched a regular campaign urging farmers in irrigated as well as rainfed areas to build ponds and reservoirs to harvest rainwater. Ponds and reservoirs are low-cost and low-maintenance water harvesting techniques that farmers can use to store rainwater during winter and use it for summer crop when irrigation water quantity drops. A ministry spokesman said, "There are also much simpler ways to benefit from rainwater during winter, such as digging holes around trees to trap rainwater or applying contour farming opposite to the runoff direction to prevent soil erosion and also trap more water." [9].

However, the prominent soil type in rainfed areas at Northern Jordan is clay characterized by low infiltration capacity and crust formation. Therefore, harvested rainwater in the planting area will be subjected to high evaporation before entering the soil profile, thus reducing the efficiency of microcatchments in increasing soil water storage. Therefore, sand-ditch technique is being investigated to enhance raped water infiltration and entry into the soil profile, thus reducing evaporation losses. The objective of this study therefore is to test sand ditch as a mean to facilitate surface infiltration, reduce evaporation, increase soil moisture storage and ultimately improve crop production in arid lands.

Sand ditch consists of constructing a trench of varying width, about 100 cm, and depth, about 80 cm, across the slope of the land, about 5 m long, to be filled with local sand and fragments of sedimentary rocks that have high permeability [10,11]. Due to its high infiltration rate, sand ditches permit rainfall and runoff water to enter soil profile rapidly, thus enhancing water infiltration and redistribution in the soil profile while limiting water evaporation. Sand ditch has been successfully implemented in an olive farm in Jordan and placed in the middle of tree rows about 3-m away from trees' location [10]. This technique improves plant growth by enhancing the root growth deeper into the soil profile seeking moisture stored around sand ditches. The growth of deep roots, compared to surface roots, improves the resilience of plants to drought in arid regions. Sand dams have been used in Kenya to conserve runoff water from streams and store it in the sand deposits for future uses [12]. Such conservation techniques may reduce the overall water balance at the catchment level, but the water stored within the farm boundary could have high beneficial return. Abu-Zreig et al. [10] found that soil moisture storage in the sand ditch area was constantly higher compared to the control area (50% higher in average) and in some cases equal to that of rainfall depth. In a controlled laboratory experiment, Saito et al. [13] found that sand ditches decreased evaporation losses by 25% compared to the control. Sand ditches is a sustainable and long-term technique that can be implemented, at the farm level, combined with microcatchment-rainwater harvesting techniques to accelerate the infiltration of collected runoff into the root zone, thus reducing water evaporation. However, limited attempts were made to monitor deep percolated water below the bottom of the sand ditch to quantify deep-water storage that can be used by summer crops. Although the experimental results are based on simple infiltration theory, experimental data are needed to assure local farmers about the efficiency of sand-ditch technique in rainwater capturing and infiltration. Therefore, the objective of this study is to examine the role of the sand-ditch technique in enhancing deep percolation of rainwater and increasing soil water capacity in the soil profile to be used for crop production in drylands.

## 2. Materials and Methods

Runoff experiments were conducted using plastic pens 50 cm long, 20 cm wide and 25 cm deep under laboratory conditions. Soil pens were supplied with a drainage layer 2 cm thick consisting of plastic mesh and cellulose papers placed at the bottom of the pens below soil. Percolated water was then collected from an outflow pipe installed at the downstream bottom end of the soil pens as shown in Figure 1. The pens were filled with soil in 3 layers of 7 cm each and compacted by tamping the whole pen 5 times for each layer then leveled with a hand tool. The runoff outflow was collected using a funnel that was fixed at the down slope end of the pens. The soil is placed at 4% slope. Figure 1 shows a photograph and a schematic diagram for the experimental setup.

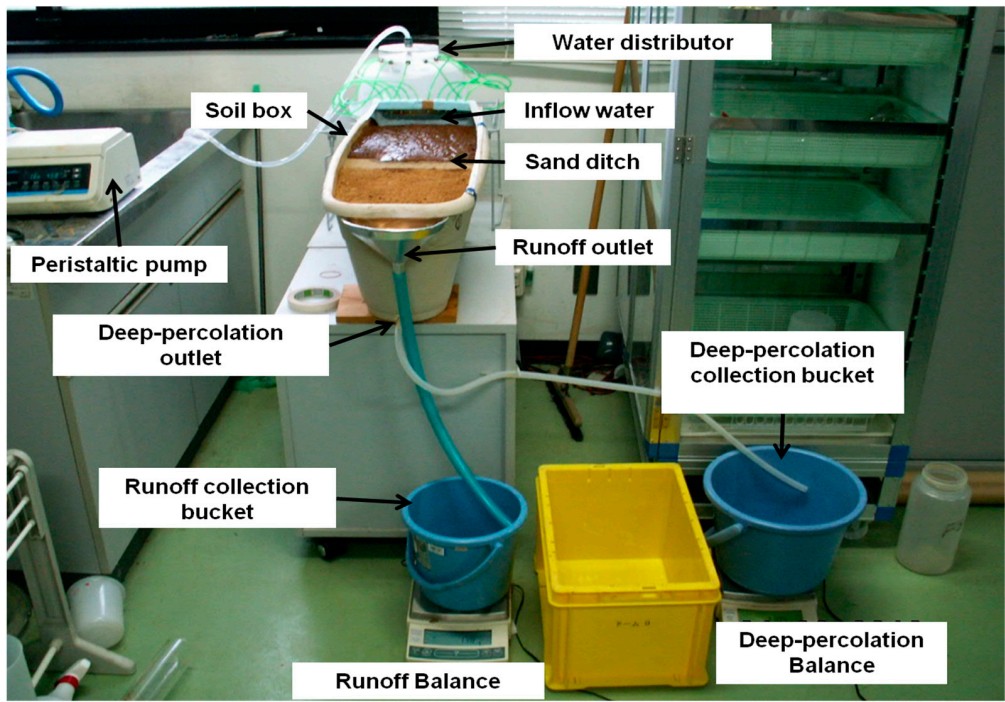

**Figure 1.** Picture for the experimental setup.

The experiments were conducted using clay soil prepared at two bulk densities equal to 1240 and 1510 kg/m$^3$, with and without sand ditches and with simulated runoff. The soil was collected from the field then sieved using 2 mm sieve openings and placed in the pens with small hand shovel. Soil saturated hydraulic conductivity and particle size distributions were measured according to standard methods [14], and the results are summarized in Table 1. The soil saturated hydraulic conductivity was measured using constant head method, while the PSD was measured using pipette methodology as described by Klute [14]. The measured saturated volumetric water content and saturated hydraulic conductivities for the low-density soil (LD) were 0.51, 3.2 × 10$^{-5}$ m/s; and 0.48, 2.7 × 10$^{-5}$ m/s for the high-density soil (HD), respectively. The sand used to fill the ditches has saturated water content of 0.42 m$^3$/m$^3$ and saturated hydraulic conductivity of 4.0 × 10$^{-4}$ m/s. To achieve a desired soil density, pens were filled with soil in three layers, and each layer was compacted with a handheld steel hammer three times. We followed the packing procedure strictly for all soil pens to maintain similar high densities for replicates. The initial soil moisture content of the soil was also measured. When a soil-pen preparation was finished, the weight of soil was recorded to calculate its dry bulk density. The volume of the soil pens used for the experiments were measured by water volume and found to be equal to 20 L. Sand ditches were positioned in the middle of pens using rectangular parallel plates of 5 cm apart and extended to the entire width of the pen. Soil in the pen and sand inside the parallel metal plate were filled simultaneously during compaction to the desired

depth. After that, the metal plates were pulled up slowly to prevent mixing of sand and soil thus creating a sand zone of 5 cm wide and extended along the width and depth of the pen.

**Table 1.** Physical properties of soils used in this research.

| Soil Name | Sand (%) | Silt (%) | Clay (%) | Saturated Water Content (m³/m³) | | Air Dry Water Content (m³/m³) | | Saturated Hydraulic Conductivity (m/s) | | Soil Classification |
|---|---|---|---|---|---|---|---|---|---|---|
| | | | | LD § | HD § | LD | HD | LD | HD | |
| Red, Tohaku | 24 | 32 | 44 | 0.51 | 0.48 | 0.03 | 0.04 | $3.2 \times 10^{-5}$ | $1.5 \times 10^{-5}$ | Clay |
| Tottori Sand | 96.1 | 0.4 | 3.5 | 0.42 | | 0.014 | | $4.0 \times 10^{-4}$ | | Dune Sand |

§ LD and HD are soil low-density and high-density, respectively.

Simulated runoff at known rates was supplied from the upslope end of the pens with a peristaltic pump using circular distributes with 10 outlets to spread runoff evenly across the width of the soil pen (Figure 1). The average inflow rates for the experiment were about 20 cm/h (20 L/h) but varied slightly due to the performance of peristaltic pump (STD = 2.7 cm/h) (2.7 L/h). We measured the inflow rates from the distributer at the beginning and at the end of each experiment by collecting inflow water volume for 10 min. Varying inflow rates among treatments should give more insight on the efficiency of sand-ditch technique under low and high runoff conditions. The experimental duration was fixed at 60 min because preliminary experiments showed that steady state condition, i.e., steady runoff and drainage rates, was achieved during the first 60 min period. Volumes of runoff and percolated water were collected in two separate plastic buckets that were placed on a digital balance for continuous and instantaneous recording of data. Cumulative weights for the runoff and percolated water volumes were recorded at various time intervals, every 1 or 2 min, during the experiments. These data also allow the calculation of deep-percolation and runoff rates in cm/h or L/h. Collection of percolated water was continued after the cutoff of inflow water until no percolated water was running from the drain outlet, which usually lasted for about 15 min. After that, the weight of saturated soil pens was recorded in order to calculate the final soil water content and subsequently the amount of soil water storage to perform water mass balance analysis.

## 2.1. Modelling of Water Infiltration in the Presence of Sand Ditches

We used numerical model WASH_2D, which solves governing equations for two-dimensional movement of water, solute and heat in soils, similar to HYDRUS-2D, using finite difference method to handle vertically layered soils with overland runoff. Although, Simunek et al. [15] recently presented a new feature of HYDRUS (2D/3D) model, which allows transient standing water formed as well as furrow or wetland, still, it cannot handle overland flow caused by runoff processes. Therefore, we modified WASH_2D model and included an overland flow component that resembles the experimental conditions presented in this research and field conditions containing runoff routing. The software is freely distributed with source code and sample cases described below under the general public license from the website of Arid Land Research Center, Tottori University [16] (http://www.alrc.tottori-u.ac.jp/fujimaki/download/WASH_2D).

### 2.1.1. Governing Equation for Water Flow

The two-dimensional water balance equation for the combined liquid and gaseous phases is given by

$$\frac{\partial \theta}{\partial t} = -\left(\frac{\partial q_{lx}}{\partial x} + \frac{\partial q_{lz}}{\partial z}\right) - \left(\frac{\partial q_{vx}}{\partial x} + \frac{\partial q_{vz}}{\partial z}\right) - S \tag{1}$$

where $\theta$ is volumetric water content (cm³/cm³), $t$ is time (s), $q_l$ is the liquid water flux (cm s$^{-1}$), $q_v$ is the water vapor flux (cm s$^{-1}$), $x$ is distance in parallel to soil surface, $z$ is depth (cm), and $S$ (cm³ s$^{-1}$/cm³) is a sink term. Subscripts $x$ and $z$ refer to direction of each flux.

The liquid water flux, $q_l$, is described using Darcy's law:

$$q_{lx} = -K\left(\frac{\partial \psi}{\partial x} - \sin \gamma\right) \tag{2a}$$

$$q_{lz} = -K\left(\frac{\partial \psi}{\partial z} - \cos \gamma\right) \tag{2b}$$

where $K$ is the hydraulic conductivity (cm s$^{-1}$), $\Psi$ is the matric potential (cm), and $\gamma$ is the angle of the slope. In this study, soil moisture was kept wet, and therefore, governing equations for vapor movement are omitted.

### 2.1.2. Theory of Overland Flow

The mass conservation equation for overland flow is given as

$$\frac{\partial h}{\partial t} = -h\frac{\partial v}{\partial x} + q_{rain} - q_{infilt} \tag{3}$$

where $h$ is height of water (cm), $t$ is time (s), $v$ is flow velocity (or flux) (cm s$^{-1}$), $q_{rain}$ is intensity of rain (cm s$^{-1}$), and $q_{infilt}$ is infiltration rate (cm s$^{-1}$). Assuming uniform flow, Manning [17] presented velocity as square root of hydraulic gradient:

$$v = \frac{h^{\frac{2}{3}}}{n}\sqrt{\frac{\partial H}{\partial x}} \tag{4}$$

where $H = h + z_g \approx h + z$, and $n$ is the roughness coefficient (s cm$^{-1/3}$). For smooth soil surface, a value of 0.0065 s cm$^{-1/3}$ (= 0.03 s m$^{-1/3}$) is recommended [18]. Under sheet flow, hydraulic gradient is dominated by gravity potential gradient (Case A). However, hydraulic gradient may also be dominated by pressure gradient caused by non-uniform rainfall intensity and/or infiltration rate even above a flat a soil surface (Case B) as shown in Figure 2.

$$\frac{\partial H}{\partial x} = \frac{\partial h}{\partial x} + \frac{\partial z_g}{\partial x} = \cos\gamma\frac{dz}{dx} \approx \frac{dz}{dx} = \tan\gamma \approx \sin\gamma \tag{5a}$$

$$\frac{\partial H}{\partial x} = \frac{\partial h}{\partial x} + \frac{\partial z_g}{\partial x} = \frac{\partial h}{\partial x} \tag{5b}$$

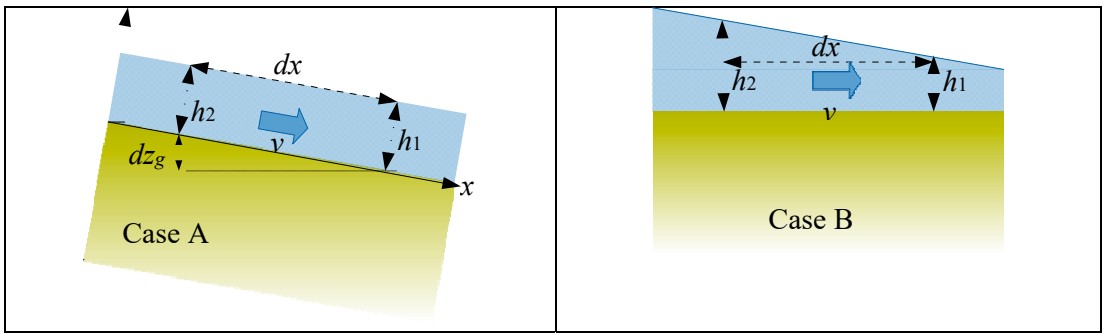

**Figure 2.** Case A is gravity gradient surface flow, and Case B is pressure gradient surface flow.

### 2.1.3. Numerical Scheme

WASH_2D solves the water flow equation with the alternating direction finite difference method based on the mass-conservative iterative scheme proposed by Celia et al. [19]. Time steps, $\Delta t$, are adjusted automatically such that, at each time step, the number of iterations is approximately five,

and the maximum change in $\ln(\psi_m)$ is smaller than prescribed value. Hysteresis is considered using a simple method by Kool and Parker [20], but it did not work under the monotonic wetting process of this study. Overland flow equation is solved with the explicit finite difference method, and ponding depth is used for calculating infiltration rate. More details on the mathematical description and solution of coupled infiltration and overland flow are shown on the Model website (http://www.alrc.tottori-u.ac.jp/fujimaki/download/WASH_2D).

## 3. Results and Discussion

We carried out eight experiments in the laboratory using clay soil with varying bulk densities and slightly varying inflow rates among replicates. The summary of those experiments and their conditions is shown in Table 2. The experimental duration lasted for 60 min until a steady runoff rate and deep-percolation rates were observed. Inflow discharges at upslope edge of the soil box, expressed in depth per time, were inflated to represent surface flow routing in the field. As shown in Table 2, there is small inflow rate variations among experiments that give insight into the nature of water infiltration and deep percolation through soil and the subsequent percentage of drainage and runoff as affected by the inflow rate among replicates. The measured parameters including runoff rates and deep-percolation rates during the experimental duration is shown in Figures 3–6. Figure 3 shows the runoff rates, and Figure 4 shows the water deep percolation rates through the drainage face for low-density soil pens. The corresponding results (runoff rates and deep-percolation rates) for control and sand ditch experiments under compacted high-density soil is shown in Figures 5 and 6, respectively.

**Table 2.** Summary of the runoff experimental conditions conducted in soil box.

| Experiment No. | Experimental Condition | Bulk Density (g/cm³) | Inflow Rate (cm/h) | Duration (min) | Inflow Volume (cm) |
|---|---|---|---|---|---|
| 1 | Control | 1.24 | 23.7 | 60 | 23.7 |
| 2 | Control | 1.24 | 20.1 | 60 | 20.1 |
| 3 | Sand ditch | 1.26 | 21.0 | 60 | 21.0 |
| 4 | Sand ditch | 1.24 | 17.0 | 60 | 17.0 |
| 5 | Control | 1.54 | 21.0 | 60 | 21.0 |
| 6 | Control | 1.48 | 16.2 | 60 | 16.2 |
| 7 | Sand ditch | 1.51 | 21.4 | 60 | 21.4 |
| 8 | Sand ditch | 1.51 | 17.1 | 60 | 17.1 |

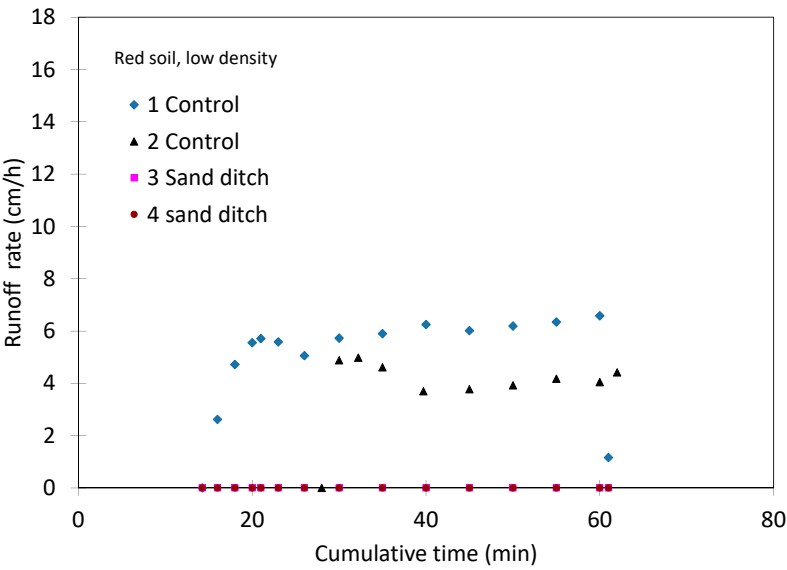

**Figure 3.** Runoff rates measured at the runoff outlet of soil boxes for low-density clay soil.

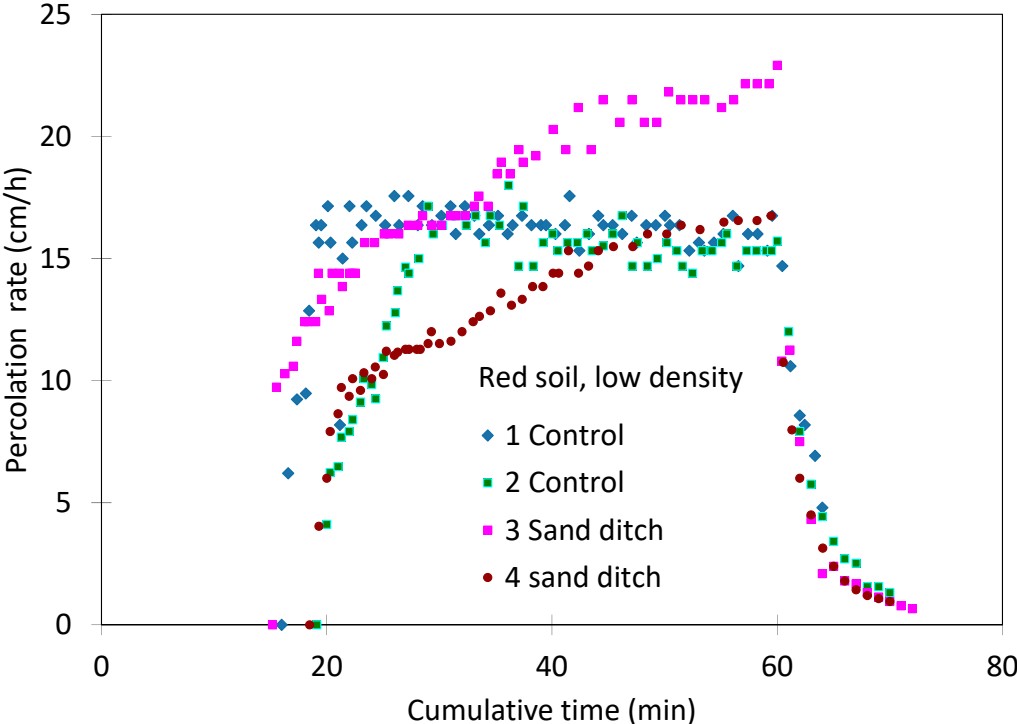

**Figure 4.** Percolated water rates measured at the drainage outlet of soil boxes for low-density clay soil.

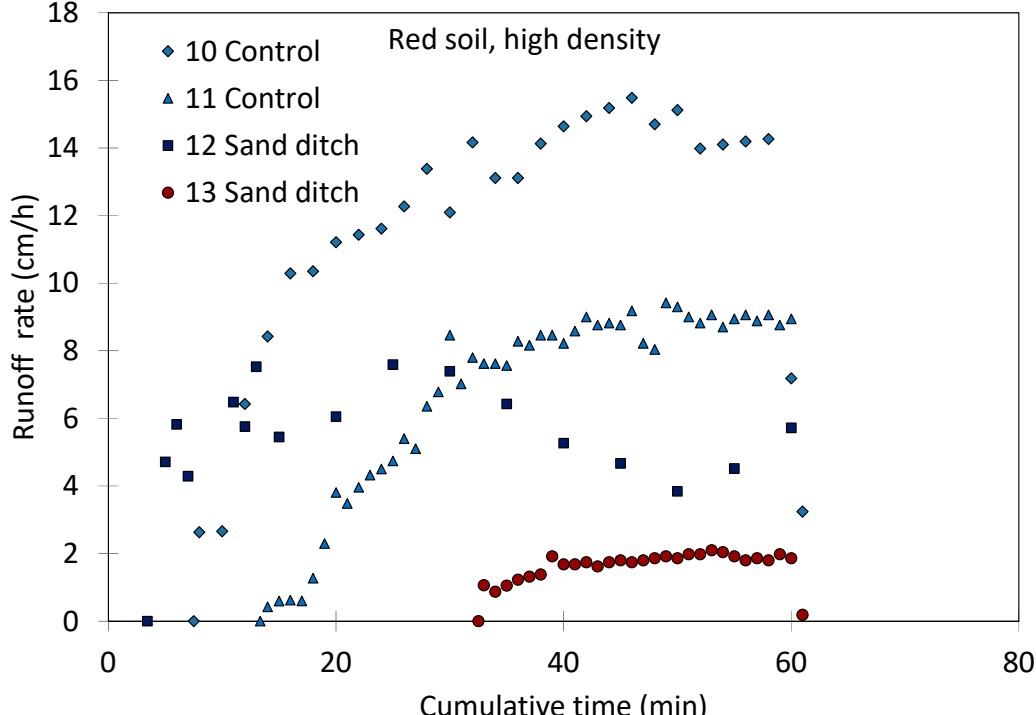

**Figure 5.** Runoff rates measured at the runoff outlet of soil boxes for compacted high-density clay soil.

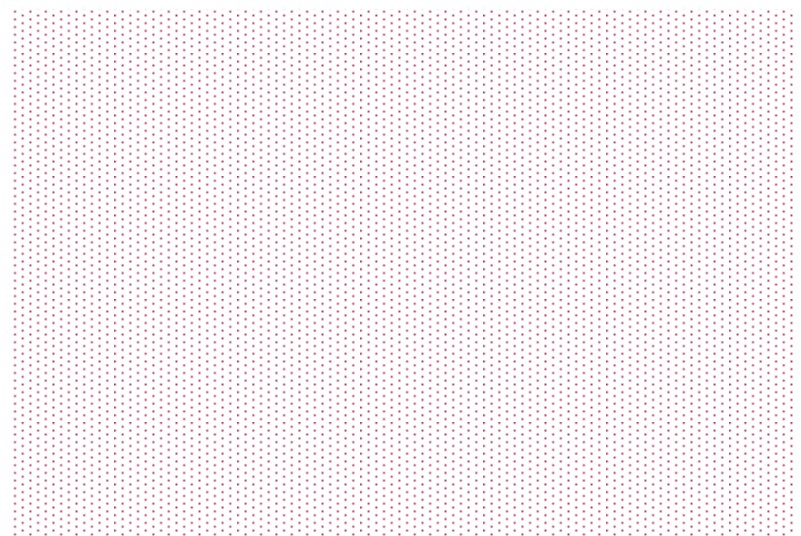

**Figure 6.** Percolated water rates of soil boxes for compacted high-density clay soil.

Runoff rates plotted in Figures 3 and 5 clearly show the influence of sand ditch on water flow. There is some variability among replications, which is attributed to the variation of inflow rates and the appearance of rills on the soil surface for some experiments. In low-density soil, runoff rates increased quickly after reaching the outlet and then became steady until cutting off the inflow water, which then seized almost immediately. In the case of high-density soil, runoff increased slowly for about 30 min and then reached steady state until the time at which inflow water was cutoff.

Installing sand ditch in low-density soil stopped the runoff entirely and forced runoff water to infiltrate. In high-density soil, runoff rates from control plots were higher than those in low density soil, as expected, but in the presence of sand ditch, runoff was reduced largely compared to control. The runoff water reached the pens' outlet at various durations and seemed to be affected by the inflow rates, soil bulk density and the presence of sand ditch (Figure 7). For control-low-density soil experiments, runoff starts after 13 and 29 min for the two replicates and reduced to 8 and 13 min for soil pens compacted with high-density soils. In the case of sand ditch experiments, runoff was eliminated in low-density pens but reached the outlet for soil pens with sand ditches in about 30 min after the start of the experiments.

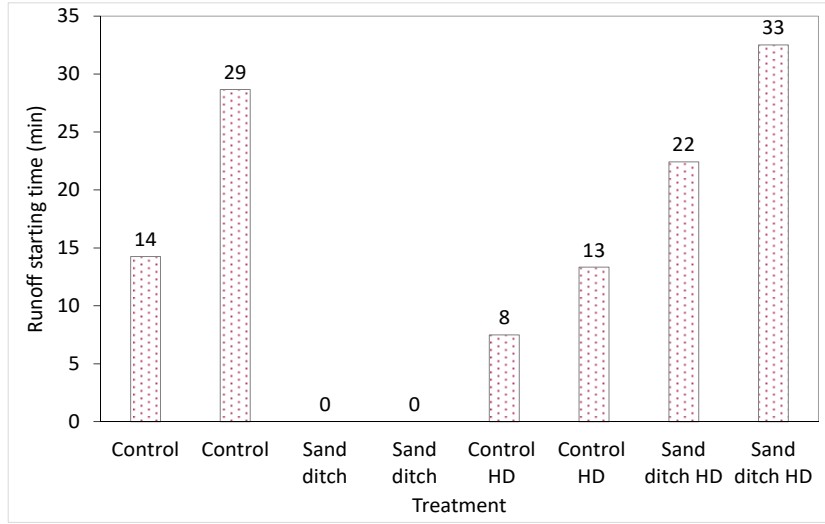

**Figure 7.** Runoff starting time at the downslope outlet of the soil box after introducing inflow water at the upslope edge for low density and compacted high-density soil.



Deep-percolation rates, measured from the drainage outlet and shown in Figures 4 and 6, follow a typical step input of steady inflow for 60 min. In general, sand ditch appears to be more effective in compacted high-density soil compared to soil at low-density. This is apparent from Figure 6 at which deep-percolation rates in sand ditches are much higher than those in the control in relation to Figure 4 where the corresponding differences of deep-percolation rates between sand ditch and control experiments are relatively small. When percolated water reached the outlet of the drain, it rose steadily for about 15 min and then became steady until the time when inflow water was cutoff. This was followed by a sharp decrease in percolated water rates until no water was running from the drainage outlet, which usually lasted for about 15 min after the cutoff of inflow water.

The arrival time of percolated water to the drainage outlet of soil pens is an indication of the speed of runoff infiltration and deep percolation into the soil profile read from Figures 4 and 6. Sand ditch seemed to enhance rapid deep percolation especially in high-density soil with an arrival time of about 16 min compared to 32 min in the control. In low-density soil, deep-percolation speed seemed to be similar in both sand ditch and control plot, about 16 min, due to high hydraulic conductivity of low-density soil.

A summary for the steady state runoff rates, deep-percolation rates and the subsequent volumes of runoff and percolated water collected at the drainage outlet are shown in Table 3. These values were estimated from Figures 3–5 when runoff rate or deep-percolation rate became steady toward the end of the experimental duration. The corresponding percentages of runoff rate, deep-percolation rate and volume to that of inflow rate and volume are calculated in Table 4 and summarized in Figure 8. It should be noted that the total sum of deep-percolation rate and runoff rate at steady state, occurring when soil water storage became constant, should be approximately equal to that of inflow rate (Table 3). The influence of inflow rate among replicates on runoff rate, deep-percolation rate and soil moisture content of soil pens can be comprehended from Table 3. Table 3 clearly shows that smaller inflow rates among replicates resulted in lower runoff at the surface outlet and higher water content in the control as well as in sand ditch pens. for example, the runoffs resulting from the control experiments operated at inflow rates of 23.7 cm/h and 20.1 cm/h (Experiments No 1 and 2) were 6.3 and 4.1 cm/h, respectively; the resultant soil water contents for those experiments were 34.5% increasing to 36.2%, respectively.

**Table 3.** Summary of steady state runoff measured at the outlet and deep-percolation rates and volume measured at the drainage face as affected by sand ditch treatment, soil type and bulk density.

| Experiment No. | Inflow Rate (cm/h) | Runoff Rate (cm/h) | Deep-Percolation Rate (cm/h) | Runoff Volume (cm$^3$) | Deep-Percolation Volume (cm$^3$) | Volumetric Water Content (%) |
|---|---|---|---|---|---|---|
| 1 | 23.7 | 6.3 | 16.2 | 4.5 | 12.4 | 34.5 |
| 2 | 20.1 | 4.1 | 15.7 | 2.3 | 10.6 | 36.2 |
| 3 | 21.0 | 0.0 | 20.9 | 0.0 | 14.2 | 34.1 |
| 4 | 17.0 | 0.0 | 16.6 | 0.0 | 17.0 | 36.1 |
| 5 | 21.0 | 14.3 | 6.6 | 10.7 | 4.2 | 30.9 |
| 6 | 16.2 | 8.8 | 7.3 | 5.4 | 3.4 | 37.2 |
| 7 | 21.4 | 5.9 | 15.5 | 5.1 | 9.8 | 32.1 |
| 8 | 17.1 | 1.9 | 15.2 | 0.8 | 9.6 | 33.8 |

**Table 4.** The ratios of steady state runoff measured at the surface outlet and deep-percolation rates and volume measured at the drainage face to the corresponding inflow rates and volumes for the experiments.

| Experiment No. | Runoff Rate Ratio (%) | Deep-Percolation Rate Ratio (%) | Runoff Volume Ratio (%) | Deep-Percolation Volume Ratio (%) |
|---|---|---|---|---|
| 1 | 26.6 | 68.4 | 18.8 | 52.2 |
| 2 | 20.6 | 78.2 | 11.7 | 52.5 |
| 3 | 0.0 | 99.7 | 0.0 | 67.7 |
| 4 | 0.0 | 97.5 | 0.0 | 57.8 |
| 5 | 68.3 | 31.4 | 51.0 | 19.8 |
| 6 | 54.4 | 44.8 | 33.1 | 21.2 |
| 7 | 27.5 | 72.3 | 23.9 | 45.9 |
| 8 | 10.9 | 89.1 | 4.6 | 56.0 |

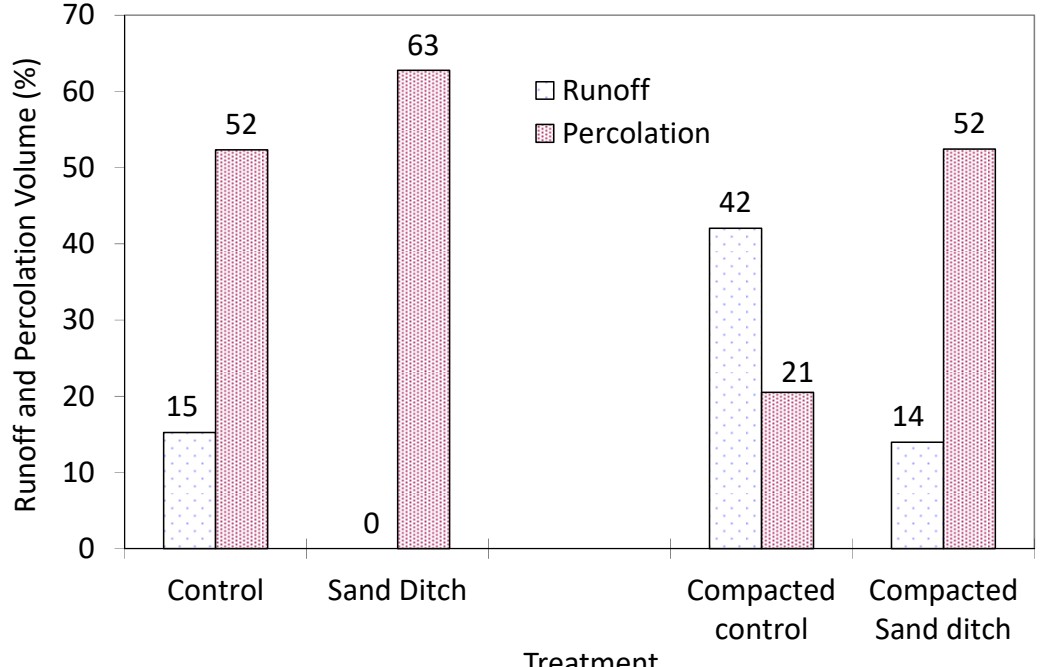

**Figure 8.** Average of runoff and deep-percolation volume as percentage of the inflow volumes for the experimental trials.

Previous research carried out by Abu-Zreig et al. [10] has also observed that installing sand ditches in the field across land slope increased soil moisture storage significantly compared to soil without sand ditches. Other experiments have shown that sand ditches installed in field plots of 20 m$^2$ reduce runoff by 30% compared to control plots [11]. However, no previous works have measured quantitatively the amount of water percolated into the soil profile below sand ditch level. The present work confirmed that installing sand ditches across land slope intercepts runoff and increases deep percolation, reaching 95% of runoff, at the present experimental conditions, thus increasing deep soil water storage and increasing the potential of groundwater recharge.

Experiments No. 5–8, shown in Table 2, were carried out using compacted soil, to have more insight on the influence of sand-ditch technique in low-hydraulic conductivity soil. The bulk density of compacted soil was 1500 kg/m$^3$ compared to 1240 kg/m$^3$ for uncompacted or loose soil resulting in 2-times lower hydraulic conductivity (Table 1). Data obtained from the control experiments confirmed that the steady state deep-percolation rates in the control for loose soil, Experiments 1 and 2, is about two times higher than that for compacted soil, Experiments No 5 and 6, shown in Table 2. Table 3 also showed that the influence of sand ditch on water deep percolation for compacted soil was more

profound than that for loose soil. Installation of sand ditches was unable to prevent runoff from reaching the outlet for compacted soil but reduced runoff by 3 times in average, from about 11.6 cm/h in the control to 3.9 cm/h in sand ditch pens. Subsequently, the average drainage rate increased from 7.0 cm/h for control pens to as high as 15.4 cm/h for experiments with sand ditches, a 120% increase. Similar increase was also observed for the volume of percolated water collected from the drainage outlet. The amount of drainage water for control experiments was about 3.8 cm in average and increased to 9.7 cm in the presence of sand ditches, 156% increase, much higher than the observed increase in the drainage volume for loose soil in the presence of sand ditch (experiments No. 1–4), which was only 36% compared to control. This analysis indicates that although sand ditches represent only 10% of the soil area, they largely improve deep percolation especially in compacted or heavy soil having low hydraulic conductivity and/or infiltration capacity.

Studies involving on-farm rainwater harvesting have demonstrated the importance of the existence of high hydraulic conductivity zones. Saito et al. [13] showed that sand ditches improved water infiltration by 30% compared to control, but his study did not involve measuring deep percolation of infiltrated water. Dages et al. [21] have also reported that localized groundwater recharge resulting from deep percolation of irrigation water contributes to about 50% of total recharge despite the fact that irrigation canals represents only 6% of the study area.

*Model Performance*

We attempted to simulate the experimental trials conducted in this research with limited success. The major errors resulted from the large runoff rate that seemed to cause model instability. The second source of instability resulted from the large differences in moisture content and soil matric potential between clay soil and sandy soil in the ditch. A large decrease in the matric potential at the clay-sand interface seemed to cause model instability and crash. Nevertheless, we successfully modified some of the experimental conditions and used the model simulation with and without sand ditches to support our experimental findings. Model simulations were performed by reducing the input runoff rate, to 3 cm/h, reducing time step during simulation and assuming relatively wet initial condition (uniform at = −200 cm). Left and right boundary conditions were kept impermeable, and the bottom one was the seepage face.

Figures 9 and 10 demonstrate the graphical representation of water deep percolation and runoff from two soil boxes with and without sand ditches. Hydraulic properties of loamy soil from Mushaqar, Jordan, and sand from Tottori, Japan, were used for matrix and sand ditch, respectively, whose hydraulic properties are shown in Figure 11. The width and height of the region are 50 cm and 25 cm, respectively. At $t$ = 0.5 h, 9.1 cm$^2$ of water runoff toward right side from the region occurred for uniform soil while 10.9 cm$^2$ of drainage occurred instead of runoff with a sand ditch. These results are qualitatively in accordance with the experimental results. Interestingly, the contour map shows that not only the sand ditch but also the soil surrounding the sand ditch is wet, and thus, storage and downward flow are facilitated there.

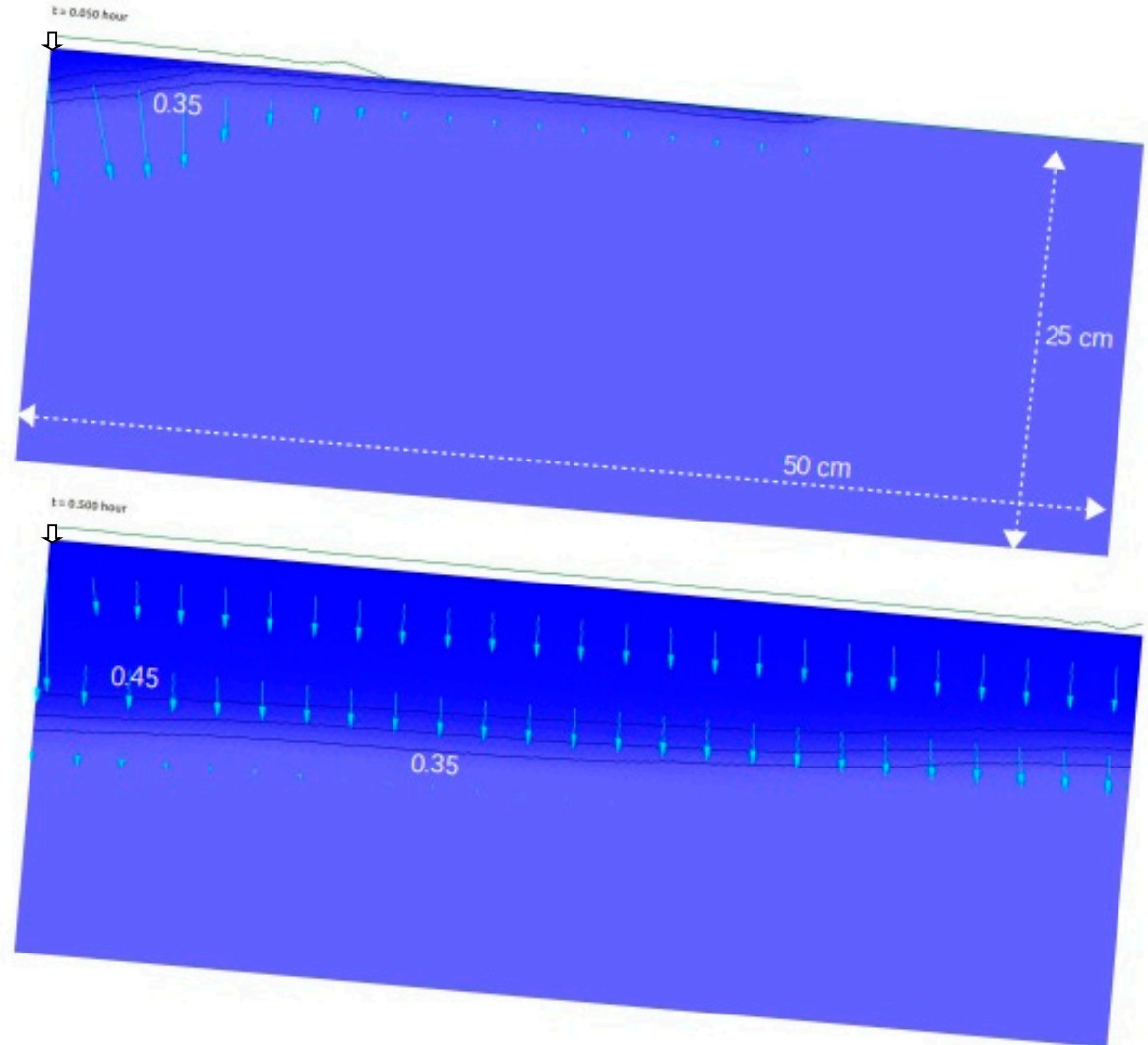

**Figure 9.** Flow simulation on loamy soil without sand ditch at *t* = 0.05 h (upper) and *t* = 0.5 h (lower) after initiation of inflow from the left upper edge at 3 cm/h (↓). Arrows show water flux with a scale of 1 (cm/h)/cm. Contour lines represent volumetric water content at 0.05 increment. Lime colored lines above the soil surface draw the height of standing water with 10 times zooming vertically.

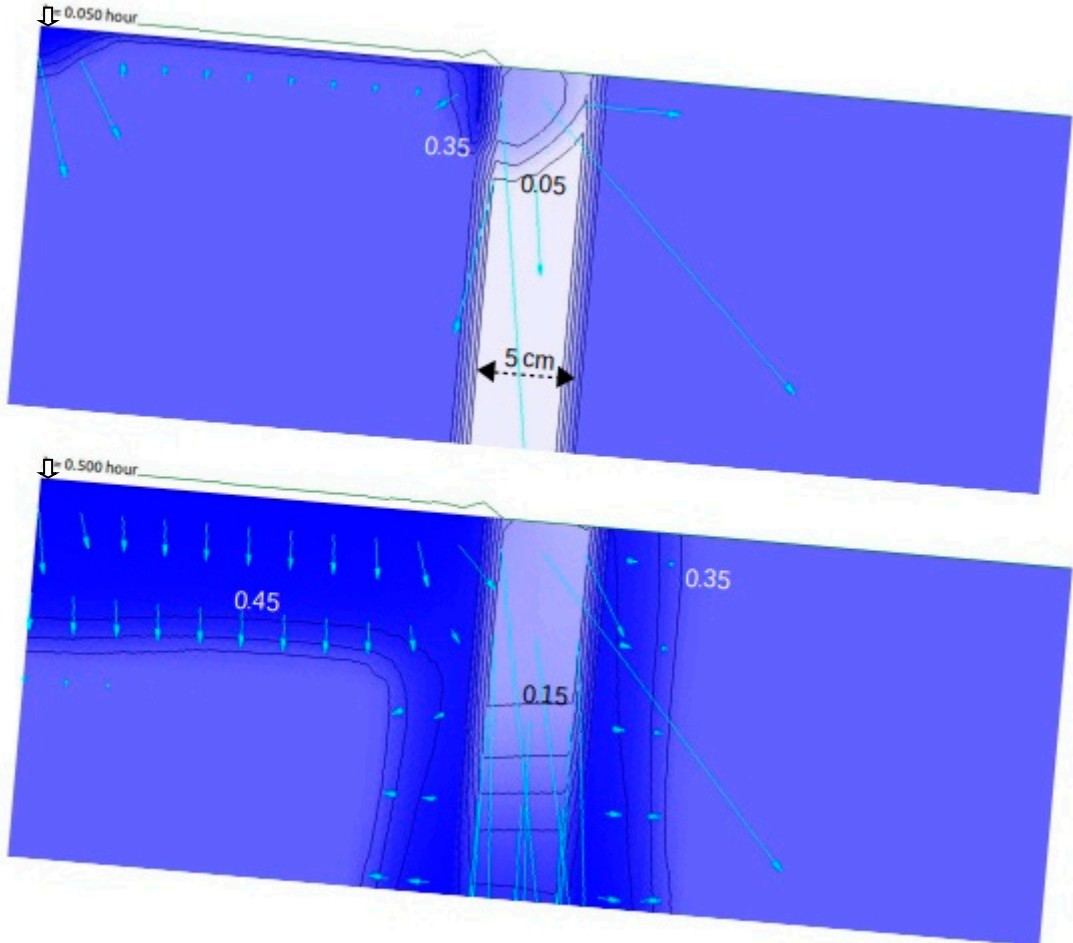

**Figure 10.** Flow simulation on loamy soil with sand ditch filled with Tottori sand at *t* = 0.05 h (upper) and *t* = 0.5 h (lower) after initiation of inflow from the left upper edge at 3 cm/h (↓).

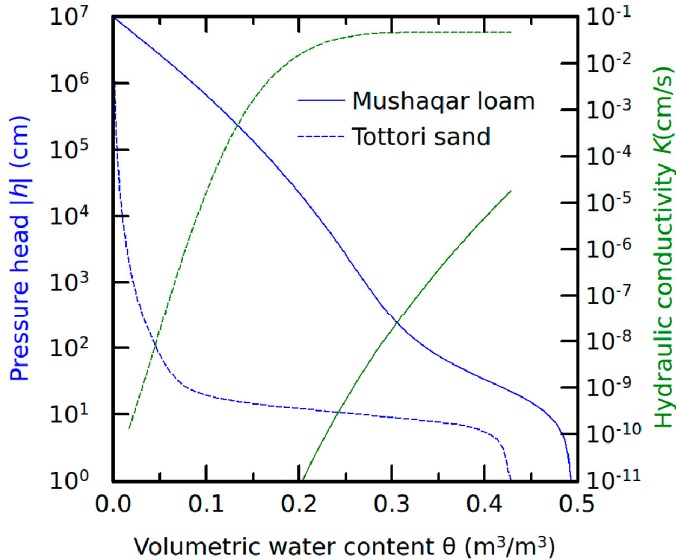

Hydraulic properties of the soils used in the simulation

**Figure 11.** Hydraulic properties of soils used in the numerical simulation.

## 4. Conclusions

This paper quantifies the effect of installing sand ditches across land slope on runoff and water infiltration and deep percolation using laboratory soil pens under simulated runoff conditions. The influence of sand ditches was also assessed using the WASH_2D model, which simulates water flow and infiltration through porous media. Due to its large hydraulic conductivity compared to surrounding soil, sand ditches are able to intercept runoff and enhance water deep percolation into the soil profile. For low density soil, sand ditches eliminate runoff causing all inflow water to percolate and subsequently appear at the drainage outlet. However, in the case of compacted high-density soil, deep-percolation rate in the control was only 37% of the inflow rate, but it increased to 80% of the inflow rate in the presence of sand ditch, a 120% increase. The volume of drainage water in sand ditch plots increased by 2.6 times compared to control plots. Water flow simulation with WASH_2D model confirms our experimental findings that sand diches increase water infiltration and deep percolation compared to control. The presence of high-permeability zones in heavy soils such as sand ditches seemed to be viable, sustainable and long-term conservation measures that can be integrated with in-farm microcatchment rainwater harvesting techniques. Microcatchments collect runoff, and sand ditches enhance deep percolation of water and increase deep soil water storage thus enhancing deep root growth of plants and improving crop production in arid and semiarid land.

**Author Contributions:** Conceptualization, M.A.-Z.; methodology, M.A.-Z., M.A.A.E.; software, H.F., M.A.-Z.; validation, M.A.-Z., H.F.; formal analysis, M.A.-Z., M.A.A.E.; data curation, M.A.-Z., M.A.A.E., H.F.; writing—original draft preparation, M.A.-Z., H.F.; writing—review and editing, M.A.-Z., H.F., M.A.A.E.; All authors have read and agreed to the published version of the manuscript.

**Funding:** This research received no external funding.

**Acknowledgments:** Thanks for Jordan University of Science and Technology and Tottori University for their support.

**Conflicts of Interest:** The authors declare no conflict of interest. The funders had no role in the design of the study; in the collection, analyses or interpretation of data; in the writing of the manuscript or in the decision to publish the results.

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
