# Peer review of "Enhancing Water Infiltration through Heavy Soils with Sand-Ditch Technique"

_water, doi:10.3390/w12051312_

Round 1

Reviewer 1 Report

General comments

Jordan belongs to the WANA (West Asia and North Africa) region, which comprises in total 22 countries. The climate of the WANA region is generally arid and semi-arid with high temperature and low rainfall. This region accounts for a major proportion of the world’s dry areas, the arable land and agrobio-biodiversity are being lost to desertification through overgrazing, deforestation, unsustainable agriculture, and industrial activities. Fresh water scarcity in the region is reaching alarming levels, while the average annual per capita renewable water supplies in WANA countries is now less than 1,500 m3, much below the world average of about 7,000 m3.

In particular, Jordan belongs to the geographical region of the West Asia region of WANA with Cyprus, Iraq, Lebanon and Syria and is considered to be one of the ten most water scarce countries in the world due to the rise of temperature and its high population growth, mainly due to the refugees from neighboring countries facing conflicts and wars, like Palestine, Iraq, Lebanon and Syria.

The manuscript examines the enhancing of the rainwater infiltration into heavy soils. The rainwater infiltration is an important strategy in arid regions to increase soil water storage and meet crop water demand. Consequently, the manuscript examines a topic of interest to several arid countries and regions worldwide.

However, it is needed major revision to be suitable for publication. Consequently, I recommend acceptance of the manuscript after major revision.

Specifically:

The authors elsewhere stated "infiltration" and elsewhere "percolation". In fact, Infiltration and percolation are two related but different processes describing the movement of moisture through soil. Τo the extent that I can know, "infiltration" is defined as the downward entry of water into the soil surface and “percolation” is the flow of water through soil. "Infiltration rate" is the rate at which a soil under specified conditions absorbs falling rain, melting snow, or surface water and "Percolation rate" although more difficult to measure directly, represents the rate at which soil moisture moves down through the soil. Regardless of the aforementioned opinions, I consider the authors have to clarify (in the main text) the two "terms". Also, they have to justify when they use the one term and when they use the other.

The manuscript has to be written according to the instructions for Authors of the WATER Journal:

  1. General check of references is needed (reference list).
  2. References must be numbered in order of appearance in the text (including table captions and figure legends).
  3. References have to be written in accordance with the Instructions for Authors of the journal, namely in the reference list the Journal’s name in abbreviation.

Specific comments

 Line 25

  1. “.Keywords:” should be “Keywords:”
  2. Please add as keyword “rainfed agriculture”

Line 29

“…but only 1.1 per cent of…” should be “…but only 1.1% of…”

Lines 29 & 30

“….of 400-600 millimeters of rain a year, and 93 percent is situated…..”

should be

“….of 400-600 mm/yr of rain, and 93% is situated…..”

Line 31

“….50-200 millimeters [1] (WWAP, 2012).” should be “….50-200 mm [1]”

COMMENT: Delete “(WWAP, 2012)”

Lines 31-33

The authors state: “Therefore, government must apply best water management options to mitigate the significant imbalance between limited supplies and growing demand across sectors especially agricultural sector.”

COMMENT: To the extent that I can know, the government of the country has long recognized that water resources are depleted very quickly. In addition, he knows in 10 from 12 watersheds the groundwater is pumped at a deficit (overexploitation). Many researchers believe the situation will get worse in the future due to climate change. Consequently, the manuscript will be more attractive to the international reader if information were added concerning the measures of the government, the plan to confront this trouble, etc. I believe a short overview (more or less) will be enough.

Line 35

“…profile [2] (Laryea, 1992). Rainfed…”

should be

“…profile [12]. Rainfed…”

Lines 43 & 44

“…..available to plants [3] (Oweis and Hachum, 2009).”

should be

“…..available to plants [15].”

Line 45

“…from runon area to planting…”

should be

“…from runoff area to planting…”

Lines 45-47

“…area for crop use. [4, 5, 6, 7] (Shadeed and Lange, 2010; Oweis et al., 1999; Abu Awwad and Shatanawi, 1997; Bruins et al., 1986) and has been used by Nabateans almost 2000 years ago [8] (Fidelibus & Bainbridge, 1994)”.

should be

“…area for crop use [17, Add Oweis et al. 2009 in the reference list, 1, 4] and has been used by Nabateans almost 2000 years ago [6]”.

Lines 48 & 49

“…only get 3-4 inches of rain a year. A microcatchment is…”

should be

“…only get 3-4 in/yr of rain. A microcatchment is…”

Lines 58 & 59

“…more water.” [9] (Hana Namrouqa, The Jordan Times, 2014).”

should be

“…more water [8].”

Lines 69 & 70

“…have high permeability [10, 11](Abu-Zreig et al., 2000; Abu-Zreig and Al 69 Tamimi, 2011).”

should be

“…have high permeability [2, 3].”

Line 73

“…future uses [12] (Lasage et al., 2008). Abu-Zreig et al. (2000) [10] found that…”

should be

“…future uses [13]. Abu-Zreig et al. [3] found that…”

Lines 75 & 76

“…experiment, Saito et al. (2003) [13] found that…”

should be

“…experiment, Saito et al. [16] found that…”

Line 90

“…with a hand a hand tool. The…”

should be

“…with a hand tool. The…”

COMMENT: “a hand” is twice.

Lines 101 & 102

The authors state: “Soil saturated hydraulic conductivity and particle size distribution were measured according to standard methods (Klute, 1986) and the results are summarized in Table 1.”

COMMENT: Please, define exactly the method used.

Line 102

“…standard methods (Klute, 1986) and…”

should be

“…standard methods [10] and…”

Line 104

“….for the high density soil. The…”

should be

“….for the high density soil, respectively. The…”

Table 1

Please, define the abbreviations “LD” and “HD”

Lines 121, 122, 132

“cm/hr” & “L/hr” should be “cm/h” & “L/h”, respectively.

COMMENT: In the International System of Units the symbol of “hour” is “h” no “hr”

Line 137

“Simunek et al. (2018) [14] presented…”

should be

“Simunek et al. [18] presented…”

Line 163

“…Manning (1891)[15]…”

should be

“…Manning [14]…”

Line 167

“…is recommended [16] (Hillel, 1998). Under…”

should be

“…is recommended [9]. Under…”

Line 171

“A)” should be “Case A”

Line 172

“B)” should be “Case B”

Line 191

“…is shown in Figures 3 to Figures 6.”

should be

“…is shown in Figures 3 to 6.”

Line 194

“…in Figures 5 and Figure 6, respectively.”

should be

“…in Figures 5 and 6, respectively.”

Tables 2 & 3, Figures 3, 4, 5 & 6, Pages 279, 280, 314, 315

“cm/hr”  should be “cm/h”

Line 253

“….shown in Figure 4 and 6, follow…”

should be

“….shown in Figures 4 and 6, follow…”

Line 256

“…the control; Versus Figure 4  where the…”

should be

“…the control in relation to Figure 4  where the…”

Line 263

“…read from Figure 4 and Figure 6. Sand…”

should be

“…read from Figures 4 and 6. Sand…”

Line 296

“…by Abu-Zreig et al., (2000) [10] have…”

should be

“…by Abu-Zreig et al. [3] have…”

Line 299

“…to control plots (Abu-Zreig and Tamimi, 2011) [11]. However,…”

should be

“…to control plots [2]. However,…”

Line 320

“…(experiment No 1 to 4) which…”

should be

“…(experiments No 1 to 4) which…”

Line 324

“…Saito et al (2003) [13] showed that…”

should be

“…Saito et al. [16] showed that…”

Line 326

“…Dages et al., (2009) [19] have…”

should be

“…Dages et al. [5] have…”

Line 337

“(uniform at ψ = -200 cm). Left…”

COMMENT: The authors use the height of water “h” no the suction “ψ”. NOTE: “ψ=-h

Please, keep the symbol “h” throughout the main text.

Figures 10 & 11

Please, in higher analysis.

Figure 12

x-asis: Please, add units

y-axis: Please, use the height of water “h” instead of suction “ψ

Line 392

“Journal of Arid Environments” should beJ Arid Environ

Lines 393 &394, 395 & 396

“Agricultural  Water Management” should be “Agric Water Manag

Line 398

Applied Geography” should be “Appl Geogr

Line 401

Journal of Hydrology” should be “J Hydrol

Line 418

Physics and Chemistry of the Earth” should be “Phys Chem Earth

Line 426

Water Science and Engineering” should be “Water Sci Eng”

Line 428

Journal of Hydrology and Hydromech

Author Response

Dear Sir,

Thank you for your time and efforts. I have attached a file addressing all your comments and suggestions.

Reviewer 2 Report

The authors use an established model (Hydrus) and experiments that can test the concepts. Therefore, the general strategy makes sense. I am not sure about the numbers used (20cm/hr?), however. That is way beyond torrential rainfall, especially in an area where rainfall can be less than 10cm/yr, unless it is enhanced greatly by surface flow routing (I assume so, but the units used make it sound like rainfall; surface flow should be an areal flux, not a linear rate, or more pertinently, a divergence of an areal flux). I also worry about the sustainability of the strategy. Where is the planting supposed to occur? How does the water get to the plants, unless the planting is in the sand ditches? How should the experiment be scaled up to agricultural practice? What effect does this strategy have on the long-term water balance of the catchment? It sounds as though it is possible that the effects are only short-term, which would probably be preferable in a scenario of climate change, where water becomes scarcer where it is already scarce (thus, no losses to storage, since it mostly delays the date in a year when near-surface storage is lost through evapotranspiration). But some general answers to these questions should be entertained, since this is a potentially important effect on Jordan's water resources.

Figure 7 and Table 3 and Table 4 were difficult to follow (they need better explanations). I expect some of the other tables should be explained better as well, but these stood out.

Author Response

(The authors gave the same response as above.)

Round 2

Reviewer 1 Report

General comments

In my opinion, the author has made an important effort to improve the quality of the manuscript. This version (ver. 2) is much better from the previous (original version). However, minor revision is needed to be suitable for publication.

Specific comments

Lines 137 & 138

“….equal to 1240  and 1510 kg/m3, with…”

should be

“….equal to 1,240  and 1,510 kg/m3, with…”

Line 223

“…and S (NOTE: Add units) is a sink…”

Line 228

The symbols “ψ” and “γ” in italic font

Line 235

“…where his height..” should be “…where h (empty) is height..”

Line 257

“….are shown on the Model website (Add website).

Lines 278 – 280

“…there is small inflow rate variations among experiments that give insight to the nature of  water infiltration and deep-percolation through soil and the subsequent percentage of drainage and  runoff as affected by inflow rate among replicates..”

should be

“….there is small inflow rate variations among experiments that give insight into the nature of water infiltration and deep-percolation through soil and the subsequent percentage of drainage and runoff as affected by the inflow rate among replicates…”

Figures 3, 4, 5, 6

y-axis: “(cm/hr)” should be (cm/h)”

Line 412

“…by Abu-Zreig et al., [10] have”… should be “…by Abu-Zreig et al. [10] have”

COMMENT: Delete “,” after the word “al.”

Line 422

“…was 1500 kg/m3 compared to 1240 kg/m3 for uncompacted…” 

should be

“…was 1,500 kg/m3 compared to 1,240 kg/m3 for uncompacted…”

Figures 10, 11

Please, in higher analysis
